# Serum Intact Fibroblast Growth Factor 23 Levels Are Negatively Associated with Bone Mineral Density in Chronic Hemodialysis Patients

**DOI:** 10.3390/jcm12041550

**Published:** 2023-02-16

**Authors:** Wen-Teng Lee, Yu-Wei Fang, Mingchih Chen, Hung-Hsiang Liou, Chung-Jen Lee, Ming-Hsien Tsai

**Affiliations:** 1Division of Nephrology, Department of Internal Medicine, Shin-Kong Wu Ho-Su Memorial Hospital, Taipei 111045, Taiwan; 2Department of Medicine, Fu-Jen Catholic University School of Medicine, Taipei 242062, Taiwan; 3Graduate Institute of Business Administration, College of Management, Fu Jen Catholic University, New Taipei City 242062, Taiwan; 4AI Development Center, Fu Jen Catholic University, New Taipei City 242062, Taiwan; 5Division of Nephrology, Department of Internal Medicine, Hsin-Jen Hospital, New Taipei City 242009, Taiwan; 6Department of Nursing, Tzu Chi University of Science and Technology, Section 2, Chien-Kuo Rd., Hualien City 970046, Taiwan

**Keywords:** hemodialysis, intact fibroblast growth factor 23, C-terminal fibroblast growth factor 23, bone mineral density, end-stage renal disease

## Abstract

(1) Background: Fibroblast growth factor 23 (FGF23) is predominantly secreted from bone and plays an important role in mineral balance in chronic kidney disease. However, the relationship between FGF23 and bone mineral density (BMD) in chronic hemodialysis (CHD) patients remains unclear. (2) Methods: This was a cross-sectional observational study that involved 43 stable outpatients on CHD. A linear regression model was used to determine risk factors for BMD. Measurements included serum hemoglobin, intact FGF23 (iFGF23), C-terminal FGF23 (cFGF23), sclerostin, Dickkopf-1, *α*-klotho, 1,25-hydroxyvitamin D, intact parathyroid hormone levels and dialysis profiles. (3) Results: Study participants had a mean age of 59.4 ± 12.3 years, and 65% were male. In the multivariable analysis, cFGF23 levels showed no significant associations with the BMD of the lumbar spine (*p* = 0.387) nor that of the femoral head (*p =* 0.430). However, iFGF23 levels showed a significant negative association with the BMD of the lumbar spine (*p =* 0.015) and that of the femoral neck (*p* = 0.037). (4) Conclusions: Among patients on CHD, higher serum iFGF23 levels, but not serum cFGF23 levels, were associated with lower BMD values of the lumbar spine and femoral neck. However, further research is required to validate our findings.

## 1. Introduction

Chronic kidney disease mineral and bone disorder (CKD–MBD) manifests as hormonal changes, mineral-handling alternations and bone metabolism as the consequences of CKD. CKD–MBD is also related with mortality and cardiovascular disease as a result of increased vascular stiffness and inflammation [1,2]. Accordingly, there is increasing interest in targeting treatment for CKD–MBD to improve clinical outcomes among patients with CKD [3]. A number of osteocyte-related biomarkers have been used to predict the incidence of CKD–MBD, including sclerostin, Dickkopf-1 (DDK1), fibroblast growth factor 23 (FGF23), bone-specific alkaline, procollagen type 1 N-terminal propeptide, osteocalcin, procollagen type 1 C-terminal propeptide, hydroxyp roline, hydroxylysine, deoxypyridinoline, pyridinoline, bone sialoprotein, osteopontin, tartrate-resistant acid phosphatase 5b, receptor activator of NF-kB ligand and osteoprotegerin [4,5]. The serum levels of sclerostin, the most widely studied bone-derived marker, increase as renal function declines [6]. Sclerostin plays a major role in CKD–MBD progression through the inhibition of Wnt signaling activity [7,8]. FGF23, a commonly used marker of bone health in CKD, is mainly manufactured by osteocytes and is also considered important in maintaining mineral balance [9,10].

FGF23 is a phosphaturic hormone that is present in vivo as intact FGF23 (iFGF23) and C-terminal FGF23 (cFGF23). The structures of iFGF23 and cFGF23 are presented in Figure 1. iFGF23 is able to bind to fibroblast growth factor receptors via its N-terminus. cFGF23 binds to the Klotho complex, leading to the downregulation of the luminal membrane sodium phosphate cotransporter [11]. Impaired FGF23 cleavage has been shown to accelerate the decline in renal function [12,13] through alterations in the balance between active and inactive FGF23 [14]. Therefore, increased serum FGF23 levels are commonly observed in patients with CKD and are reported to be associated with vascular calcification [15,16], renal anemia [17,18], all-cause mortality [19,20], CKD progression [21], hypophosphatemia-related bone disease [22,23] and decreased bone mineral density (BMD) [24,25]. BMD is defined as the inorganic mineral content of bone. Dual-energy X-ray absorptiometry (DXA) measures BMD, and BMD can be used to diagnose osteoporosis [26]. Previous prospective studies have demonstrated that lower BMD predicts incident fractures in CKD stage 3a–5 patients [27].

However, studies examining the association between FGF23 and BMD levels in patients receiving hemodialysis have reported inconsistent results, with some studies reporting a significantly negative association [28,29,30,31], while others have reached alternative conclusions [32,33,34]. The association between serum FGF23 levels and BMD in CHD patients remains unclear, as many studies have measured a single type of FGF23 without adjusting for a sufficient number of variables. Accordingly, we hypothesized that asynchronous increases in serum iFGF23 and cFGF23 levels in advanced CKD patients may lead to different forms of FGF23 with varying predictive power for clinical outcomes in dialysis patients. We, therefore, conducted the present study to compare the utility of serum levels of FGF23 (iFGF23 or cFGF23) as predictors of BMD loss in patients receiving CHD with adjustment for known bone-related factors.

## 2. Experimental Section

### 2.1. Study Population

The present cross-sectional observational study comprised a total of 43 CHD patients from an outpatient clinic enrolled at the nephrology department of Shin-Kong Wu Ho-Su Memorial Hospital Medical Center. This study was designed to evaluate the association between bone-related serological markers and clinical outcomes in patients receiving CHD. Study inclusion criteria were age ≥20 and <85 years and receiving hemodialysis for more than 3 months. Exclusion criteria were myocardial infarction or stroke within 180 days prior to enrollment, hospitalized due to an infection within 90 days prior to enrollment, body mass index less than 17 kg/m^2^ and history of malignancy. Additionally, participants with unavailable BMD data were excluded. A total of 43 participants were involved in the study analysis.

In accordance with the principles of the Declaration of Helsinki, this study was performed properly and approved by the Ethics Committee of Shin-Kong Wu Ho-Su Memorial Hospital (No. 20160802R). All patients gave their written informed consent.

### 2.2. Clinical and Laboratory Data Collection

Demographic data, comorbidities and vital signs were recorded at the time of enrollment. Venous blood samples were acquired after a fast of at least 8 h. The samples were sequentially centrifuged, separated into aliquots and then stored at −20 °C for future batch analyses prior to the initiation of routine hemodialysis. Standard commercial assays and an automated test machine (Beckman, Lane Cove, NSW, AU; www.beckmancoulter.com, accessed on 10 January 2023) were adopted for serological measurements. Measured serum parameters included glucose, hemoglobin, blood urea nitrogen, creatinine, sodium, potassium, calcium, phosphate, albumin, alkaline phosphate, total cholesterol, triglyceride and iron profiles. A Roche Elecsys assay (Roche Diagnostics, Penzberg, Germany; www.roche.com, accessed on 10 January 2023) was used for serum intact parathyroid hormone (iPTH) level measurement, while Chemiluminescent immunoassay (ELISA) kits were used for serum level detection of 1, 25-dihydroxyvitamin D (IDS, Boldon Colliery, UK; www.idsplc.com, accessed on 10 January 2023), sclerostin (Biomedica, Vienna, Austria; www.bmgrp.com, accessed on 10 January 2023), dickopf-1 (Biomedica, Vienna, Austria; www.bmgrp.com, accessed on 10 January 2023) and α-klotho (IBL America, Minneapolis, MN, USA; www.ibl-america.com, accessed on 10 January 2023). FGF23 was measured using a 2-site enzyme-linked immunosorbent assay that detects two epitopes in the carboxyl-terminal portion of FGF23 (C-terminal form, cFGF23; Quidel, San Diego, CA, USA; www.quidel.com, accessed on 10 January 2023) and epitopes within the amino-terminal and carboxyl-terminal portions of FGF23 (intact form, iFGF23; Quidel, San Diego, CA, USA; www.quidel.com, accessed on 10 January 2023). Urea kinetics (parameter Kt/V) and cardiothoracic ratio (the ratio of maximum horizontal cardiac diameter to maximum horizontal thoracic diameter measured on a plain posterior–anterior chest radiograph) were also measured. The mechanisms of iFGF23 and cFGF23 binding assays are presented in Figure 2.

### 2.3. Bone Mineral Densitometry

Lumbar spine (L1–L4) and femoral neck (bilateral) BMD (in g/cm^2^) values were measured with DXA using a lunar prodigy model (GE healthcare, Madison, WI, USA). The model is based on the standard protocols reported in the manufacturer’s instructions. The following sites were examined: lumbar spine (vertebrae L1–L4) and the femoral necks of both legs. BMD presented as g/cm^2^, T-scores and z-scores was classified according to criteria established by the World Health Organization.

### 2.4. Statistical Analyses

In the present study, continuous data are presented as means ± standard deviations or medians (interquartile ranges (IQRs)) if data violate the normal distribution assumption. Categorical data are presented as numbers with proportions (%). Spearman’s rank correlation coefficient was used to examine the correlation between two variables. A simple linear regression model was adopted to identify risk factors for low BMD.

Moreover, we used multivariable regression models to evaluate the independent relationship between FGF23 and BMD levels. Log transformation was applied to some factors to approximate normal distribution. Additionally, we used a separate regression model to model changes in BMD as a function of iFGF23 and cFGF23 using a modified stepwise procedure with three modeling steps. Model 1 was adjusted for age and gender, and model 2 further involved DM, CVD and BMI. Laboratory variables significantly associated with BMD of either lumbar spine or femoral neck in the crude analysis (*p* < 0.2) were rolled into model 3. Statistical significance was considered if the *p*-value was less than 0.05. All statistical analyses were performed using SAS for Windows, version 9.3 (SAS Institute Inc., Cary, NC, USA).

## 3. Results

### 3.1. Study Population Characteristics

The present study comprised 43 stable patients receiving hemodialysis with a mean age of 59.4 ± 12.3 years and a median duration of dialysis of 5.4 (IQR, 2.4–11.1) years. Overall, 65% (*n* = 28) of participants were male; further, 54% (*n* = 23) had been diagnosed with diabetes, and 27% (*n* = 16) had a history of coronary artery disease. Serum iFGF23 (median, 673 pg/mL; IQR, 341–1100), cFGF23 (median, 1141 RU/mL; IQR, 835–1684) and hemoglobin (mean, 10.5 ± 1.0 g/dL) levels are presented in Table 1. Table 1 also presents other demographic and biochemical characteristics. Data regarding the distribution of the BMD of the lumbar spine and femoral neck are presented in Figure 3A,B, indicating the near-normal distribution according to their shapes. Moreover, right-skewed distributions were found for both iFGF23 and cFGF23 (Figure 3C,D).

### 3.2. Determinants of BMD of Lumbar Spine in Patients on CHD

Table 2 presents the results of our generalized mixed linear regression analysis for the association between the BMD of the lumbar spine and various parameters in patients receiving CHD. The results of the crude analysis indicate that body mass index (estimate: 0.03; 95% CI: 0.01, 0.04), ln(iFGF23) (estimate: −0.009; 95% CI: −0.15, −0.02), ln(Sclerostin) (estimate: 0.14; 95% CI: 0.03, 0.24) and Kt/V values (estimate: −0.49; 95% CI: −0.84, −0.14) were significantly associated with the BMD of the lumbar spine. A significant negative association was observed between serum iFGF23 levels and the BMD of the lumbar spine (*p* = 0.015, r = −0.369; Figure 4A) but not that of the femoral neck (Figure 4B) before multivariable model adjustment.

Stepwise multivariable model adjustment, presented in Table 3, indicated that ln(iFGF23) was significantly associated with the BMD of the lumbar spine across all models. The estimated effect of every unit of ln(iFGF23) on the BMD levels of the lumbar spine was −0.09 (95% CI: −0.15, −0.02) in model 1 (following adjustment for demographic characteristics), −0.11 (95% CI: −0.18, −0.05) in model 2 (following further adjustment for comorbidities) and −0.11 (95% CI: −0.16, −0.05) in model 3 (following further adjustment for laboratory parameters). However, such significant association between ln(cFGF23) and the BMD of the lumbar spine was not observed in all three models.

### 3.3. Determinants of BMD of Femoral Neck in Patients on CHD

The associations between the BMD of the femoral neck and various parameters in patients receiving CHD are shown in Table 2. Age (estimate: −0.05; 95% CI, −0.09, −0.01), body mass index (estimate: 0.02; 95% CI: 0.01, 0.04), creatinine levels (estimate: 0.04; 95% CI: 0.01, 0.06), albumin levels (estimate: 0.28; 95% CI: 0.13, 0.46) and Kt/V values (estimate: −0.38; 95% CI: −0.64, −0.11), but not ln(iFGF23) (estimate: −0.03; 95% CI: −0.09, 0.03), were significantly associated with the BMD of the femoral neck in the crude analysis. Moreover, no significant associations between serum iFGF23 levels and the BMD of the femoral neck were observed (Figure 4C,D).

As shown in Table 3, the results of the stepwise multivariable model adjustment indicate that ln(iFGF23), but not ln(cFGF23), was significantly associated with the BMD of the femoral neck from model 1 to model 3. The estimated effect of every unit of ln(iFGF23) on the BMD of the femoral neck was −0.06 (95% CI: −0.12, −0.01) in model 1 (following adjustment for demographic characteristics), −0.06 (95% CI: −0.12, −0.01) in model 2 (following further adjustment for comorbidities), and −0.06 (95% CI: −0.12, −0.003) in model 3 (following further adjustment for laboratory parameters).

### 3.4. Correction between Urea Kinetic (Kt/V) and Serum FGF23 in Patients on CHD

Figure 5 presents the correction between urea kinetic (Kt/V) and serum FGF23. A significant negative association was observed between Kt/V and serum iFGF23 levels (*p* = 0.047, r = −0.304; Figure 5C). However, Kt/V did not show any statically significant association with ln(iFGF23) (*p* = 0.366, r = −0.141; Figure 5A), ln(cFGF23) (*p* = 0.886, r = 0.022; Figure 5B) or cFGF23 (*p* = 0.958, r = 0.008; Figure 5D).

## 4. Discussion

The present cross-sectional observational study investigated the association between serum iFGF23 levels and BMD in patients receiving CHD. The results of the present investigation indicate that the serum levels of cFGF23 are not associated with BMD in patients receiving dialysis. This finding increases our understanding of the association between serum FGF23 levels and BMD by demonstrating that serum iFGF23 levels may have greater clinical relevance than serum cFGF23 levels in patients receiving CHD.

Serum FGF23 levels are rarely measured in routine clinical practice. Two commercial kits are available for measuring FGF23 levels in human plasma (iFGF23 and cFGF23). iFGF23 (but not cFGF23) has considerable diurnal variations and is highly influenced by ex vivo proteolytic degradation, thereby limiting the clinical application of cFGF23 measurements. Moreover, cFGF23 has greater bioavailability and stability than iFGF23 in vitro [35,36]. Measured serum levels of iFGF23 and cFGF23 can differ from expected values in certain diseases. Higher serum iFGF23 levels may be observed in X-linked hypophosphatemia, autosomal dominant hypophosphatemic rickets and tumor-induced osteomalacia. Serum iFGF23 levels may be normal in iron deficiency, although higher serum levels of cFGF23 may be observed, as FGF23 synthesis and cleavage are both upregulated in iron-deficient states [14]. Accordingly, we hypothesized that the different forms of FGF23 may have differing clinical relevance in advanced CKD patients. Considering that serum FGF23 can be removed by means of hemodialysis and that the molecular weight of cFGF23 is lower than that of iFGF23, Kt/V was expected to be more negatively correlated with cFGF23 and to possible affect the statistical result of the association with BMD. Our study, however, showed that statistical significance was only found in the negative correlation between Kt/V and iFGF23 (Figure 5C) and not cFGF23. These surprising results could indicate that iFGF23 could be more physically representative in CHD patients.

Previous studies examining the association between BMD and serum FGF23 levels have reported inconsistent results among CHD patients (Table 4). Three of those previous studies reported no significant associations between serum FGF23 levels and BMD. Ureña et al. posited that increased serum cFGF23 concentrations in patients receiving hemodialysis have no effects on BMD [32]. Park et al. concluded that serum iFGF23 levels were positively correlated with serum phosphate levels but not with BMD [33]. Zheng et al. similarly concluded that serum iFGF23 levels were not correlated with BMD, osteopenia or osteoporosis [34]. However, the other four studies concluded that serum FGF23 levels and BMD presented a significant association. Malluche et al. noted that increased serum cFGF23 levels were predictive of bone loss according to DXA measurements of the BMD of the lumbar spine at baseline and after one year [28]. Wu et al. posited that increased serum iFGF23 levels predict the development of osteopenia and osteoporosis in patients on CHD, but the researchers did not show the relationship between iFGF23 and BMD [29]. The findings of the studies by Bouksila et al. [30] and Slouma et al. [31] also support an association between serum iFGF23 levels, and osteopenia and osteoporosis, but the researchers also failed to mention BMD.

In the present study, a negative association was observed between serum iFGF23 levels and the BMD of the lumbar spine (Figure 4A). The negative association with BMD also involved the femoral neck after adjustment for all three multivariable models, including age, gender, diabetes mellitus, cardiovascular disease, body mass index and important bone-related serum factors (Table 3). Our results corroborate the findings of previous studies [29,30,31] in demonstrating a significant association between serum iFGF23 levels and BMD. We believe our study to be the first one to demonstrate that the serum levels of iFGF23, but not those of cFGF23, are associated with BMD loss. In patients with ESRD, circulating levels of cFGF23 and iFGF23 have been shown to progressively increase. As CKD progresses to ESRD, the ratio of iFGF23 to cFGF23 also increases [13,36]. This may explain the lack of a significant association between serum cFGF23 levels and BMD in patients receiving CHD. A study by Shimada et al. concluded that virtually all circulating FGF23 in chronic peritoneal dialysis patients is intact and biologically active [13], a characteristic to similar to that of our study population of chronic hemodialysis patients. However, the study did not show the association between iFGF23 and BMD in chronic peritoneal dialysis. Further studies are required to fully elucidate the pathophysiological mechanisms underlying the association between circulating FGF23 and BMD in advanced-stage chronic kidney disease patients.

There were some limitations to the present study. First, a causal relationship between iFGF23 and BMD levels could not be inferred due to the study using a cross-sectional observational design. Malluche et al. used a prospective cohort study design to assess causation and demonstrate that higher serum FGF23 levels predict BMD loss [28]. Second, the present study had a relatively small sample size. A generalized linear mixed model with repeated measurements may overcome the small sample size limitation and increase statistical power. However, repeated measurements could incorporate too many confounding factors, which might lead to over-adjustment and decreased generalizability [37]. Moreover, previous studies have reported findings similar to those of the present study [28,29,30,31,32,33,34], indicating that the association between serum iFGF23 levels and BMD may have a pathophysiological basis. In addition, the measurement of the BMD of the lumbar spine in patients undergoing hemodialysis may not be as accurate as that of the BMD of the femoral neck due to the BMD of the lumbar spine possibly being affected by ectopic calcification or calcification of the vascular system [38,39,40]. Finally, the association between serum iFGF23 levels and BMD may be affected by in vitro bioactivity or dietary phosphate restrictions; this uncertainty represents a further limitation to the present study [41,42]. We did not restrict the dietary phosphate intake of the study participants, which may have introduced bias into our study analysis. Aside from the above limitations, the present study also had two major strengths. First, important bone-related markers, including sclerostin, DDK1, α-klotho, iPTH and 1,25(OH)D_3_, were measured and incorporated into the study analyses. Second, the negative association between serum iFGF23 levels and BMD observed in the present study was strong and remained after stepwise adjustments, indicating strong inference.

## 5. Conclusions

Serum levels of iFGF23 but not cFGF23 are significantly and negatively associated with BMD in patients receiving CHD. Therefore, the measurement of the serum levels of iFGF23 rather than cFGF23 should be considered when evaluating the effect of serum FGF23 levels on clinical outcomes in patients receiving CHD.

## Figures and Tables

**Figure 1 jcm-12-01550-f001:**
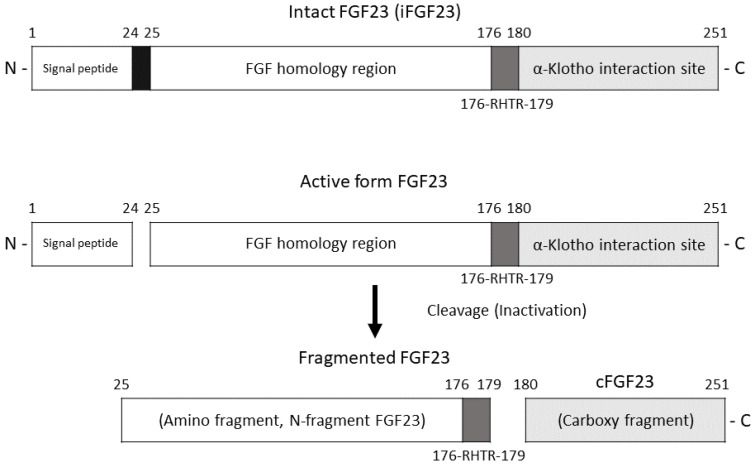
Structures of iFGF23 and cFGF23. iFGF23 protein is cleaved into an amino fragment (N-fragment FGF23, 18 kDa) and a carboxy-terminal fragment (cFGF23, 12 kDa). Note that 176-RHTR-179 represents the amino sequence of 176Arg-His-Thr-Arg179, which is the furin cleavage site.

**Figure 2 jcm-12-01550-f002:**
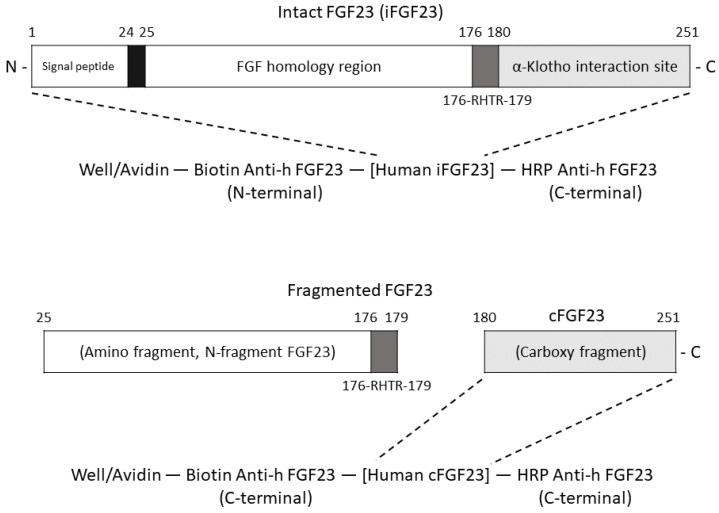
Mechanisms of iFGF23 and cFGF23 binding assays. Both iFGF23 and cFGF23 ELISA Kits are two-site enzyme-linked immunosorbent assays. Note that iFGF23 and cFGF23 are detected by two different affinity-purified goat polyclonal antibodies. One of them is biotinylated (Biotin Anti-h FGF23), able to bind with the epitopes from FGF23. The other antibody is conjugated with enzyme horseradish peroxidase (HRP Anti-h FGF23), playing a role in FGF23 detection. The biotinylated antibody, iFGF23/cFGF23 and HRP are formed into a “sandwich” complex.

**Figure 3 jcm-12-01550-f003:**
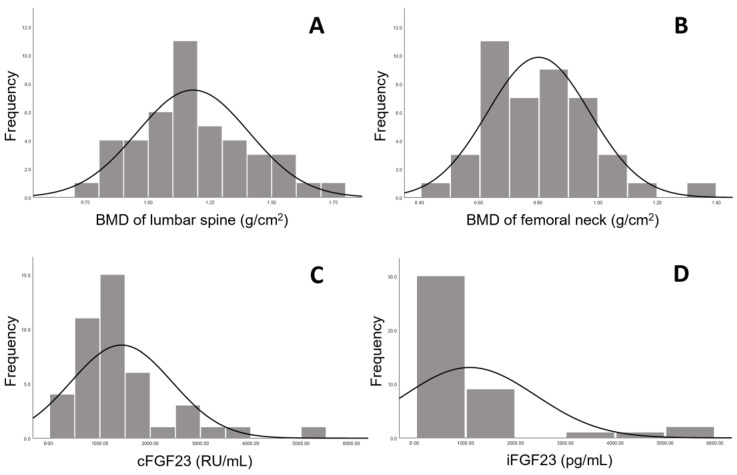
Bone mineral density distributions of (**A**) lumbar spine (L1–L4) and (**B**) femoral neck (bilateral). Distributions of (**C**) c-terminal FGF23 (cFGF23) and (**D**) intact FGF23 (iFGF23).

**Figure 4 jcm-12-01550-f004:**
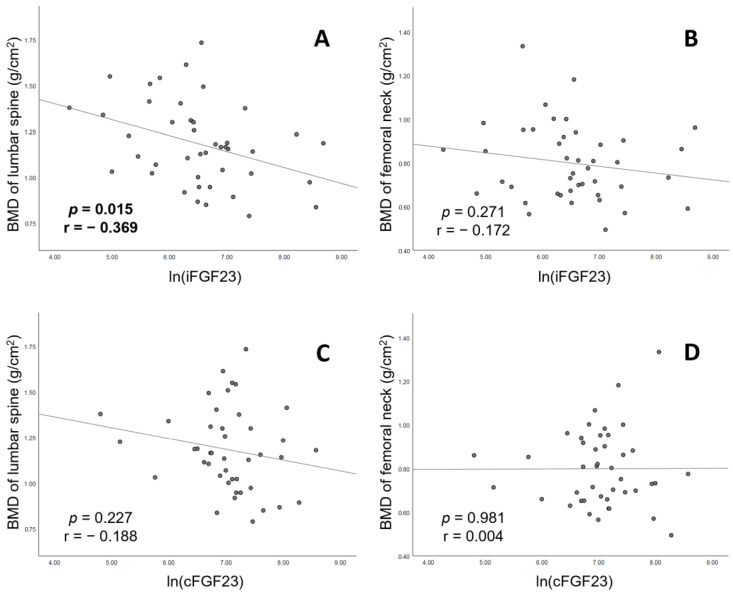
Ln(iFGF23) versus bone mineral density (BMD) levels of (**A**) lumbar spine and (**B**) femoral neck (bilateral). Ln(cFGF23) versus BMD levels of (**C**) lumbar spine and (**D**) femoral neck. The regression lines in the figure are based on the least mean squares method.

**Figure 5 jcm-12-01550-f005:**
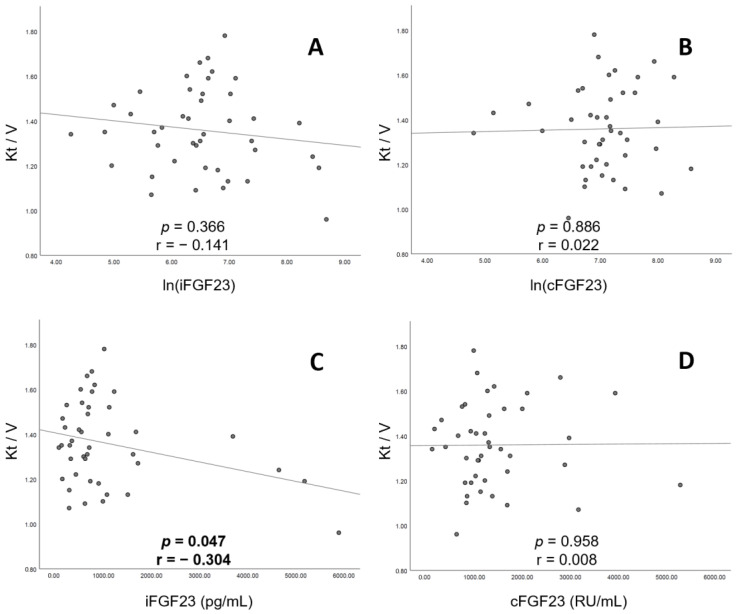
Correlations between Kt/V levels and (**A**) ln(iFGF23), (**B**) ln(cFGF23), (**C**) cFGF23 levels and (**D**) iFGF23 levels. The regression lines in the figure are based on the least mean squares method. Abbreviations: ln, natural log transformation; iFGF23, intact fibroblast growth factor 23; cFGF23, c-terminal fibroblast growth factor 23; ln, natural log transformation; Kt/V, urea kinetic.

**Table 1 jcm-12-01550-t001:** Characteristics of the study population.

Characteristic	Value (*n* = 43)
Age (years)	59.4 ± 12.3
Male (N (%))	28 (65.1)
Duration of dialysis (year)	5.4 (2.4, 11.1)
Diabetes mellitus (N (%))	23 (53.5)
Cardiovascular disease (N (%))	16 (27)
Body mass index (kg/m^2^)	23.1 ± 3.4
iFGF23 (pg/mL)	673 (341, 1100)
cFGF23 (RU/mL)	1141 (835, 1684)
DDK1 (pmol/L)	3.34 (1.64, 6.41)
Sclerostin (pmol/L)	167 ± 71
α-Klotho (pg/mL)	102 (75, 161)
1,25(OH)_2_D (pmol/L)	5.44 (2.35, 10.67)
Blood nitrogen (mg/dL)	75.5 ± 16.3
Creatinine (mg/dL)	10.71 ± 2.0
Sodium (meq/L)	138 ± 3
Potassium (meq/L)	4.6 ± 0.6
Calcium (mg/dL)	9.3 ± 1.3
Phosphate (mg/dL)	5.1 ± 1.2
Albumin (g/dL)	4.1 ± 0.3
Triglyceride (mg/dL)	106 (69, 186)
Cholesterol level (mg/dL)	166 ± 38
Kt/V	1.36 ± 0.19
Hemoglobin (g/dL)	10.5 ±1.0
iPTH (pg/mL)	214 (89, 426)
Alk-P (U/L)	68 (49, 89)
Transferrin saturation (%)	32.6 ± 12
Cardiothoracic ratio (%)	47.3 ± 5.5
Lumbar spine BMD (L1–L4) (g/cm^3^)	1.18 ± 0.23
Femoral neck BMD (bilateral) (g/cm^3^)	0.80 ± 0.17

Abbreviations: iFGF23, intact fibroblast growth factor 23; cFGF23, c-terminal fibroblast growth factor 23; DKK1, Dickkopf-1; Kt/V, urea kinetics; iPTH, intact parathyroid hormone; Alk-P, alkaline phosphatase; BMD, bone marrow density.

**Table 2 jcm-12-01550-t002:** Risk analysis of the bone mineral density in patients on chronic hemodialysis.

Parameter	BMD of Lumbar Spine		BMD of Femoral Neck	
	Estimate (95% CI)	*p*	Estimate (95% CI)	*p*
Age (per 10 years)	0.01 (−0.05, 0.07)	0.794	−0.05 (−0.09, −0.01)	**0.030**
Male (vs. female)	0.06 (−0.09, 0.20)	0.431	0.03 (−0.09, 0.14)	0.644
Duration of HD (per 10 years)	−0.03 (−0.14, 0.07)	0.524	−0.05 (−0.13, 0.03)	0.208
Diabetes mellitus (vs. none)	0.09 (−0.05, 0.23)	0.204	0.02 (−0.09, 0.13)	0.723
CVD (vs. none)	0.03 (−0.12, 0.17)	0.726	0.01 (−0.11, 0.12)	0.904
Body mass index (per 1 kg/m^2^)	0.03 (0.01, 0.04)	**0.016**	0.02 (0.01, 0.04)	**0.010**
ln(iFGF23) (per 1 unit)	−0.09 (−0.16, −0.02)	**0.015**	−0.03 (−0.09, 0.03)	0.271
ln(cFGF23) (per 1 unit)	−0.06 (−0.16, 0.04)	0.227	0.00 (−0.08, 0.08)	0.981
ln(DDK1) (per 1 unit)	−0.01 (−0.08, 0.06)	0.772	−0.01 (−0.06, 0.04)	0.720
ln(Sclerostin) (per 1 unit)	0.14 (0.03, 0.24)	**0.012**	0.05 (−0.03, 0.14)	0.224
ln(α-Klotho) (per 1 unit)	0.01 (−0.08, 0.10)	0.864	−0.00 (−0.07, 0.07)	0.912
ln(vitD) (per 1 unit)	−0.01 (−0.05, 0.04)	0.808	0.02 (−0.01, 0.05)	0.212
Creatinine (per 1 mg/dL)	0.01 (−0.02, 0.05)	0.414	0.04 (0.01, 0.06)	**0.004**
Albumin level (per 1 g/dL)	0.14 (−0.12, 0.40)	0.268	0.28 (0.13, 0.46)	**0.004**
ln(Triglyceride) (per 1 unit)	0.08 (−0.03, 0.19)	0.140	0.08 (−0.00, 0.16)	0.059
Cholesterol (per 10 mg/dL)	−0.00 (−0.00, 0.00)	0.336	−0.00 (−0.02, 0.01)	0.264
Kt/V (per 1 unit)	−0.49 (−0.84, −0.14)	**0.007**	−0.38 (−0.64, −0.11)	**0.007**
Hemoglobin (per 1 g/dL)	0.01 (−0.06, 0.09)	0.747	−0.01 (−0.06, 0.05)	0.795
ln(iPTH) (per 1 unit)	−0.03 (−0.08, 0.02)	0.275	−0.01 (−0.05, 0.03)	0.721
ln(Alk-P) (per 1 unit)	−0.05 (−0.19, 0.09)	0.463	0.02 (−0.09, 0.13)	0.698
Calcium (per 1 mg/dL)	−0.02 (−0.07, 0.04)	0.497	−0.02 (−0.06, 0.02)	0.413
Phosphate (per 1 mg/dL)	0.01 (−0.05, 0.08)	0.689	0.03 (−0.02, 0.08)	0.242
Cardiothoracic ratio (per 1%)	−0.25 (−1.55, 1.04)	0.692	−0.17 (−1.15, 0.82)	0.735

Abbreviations: HD, hemodialysis; CVD, cardiovascular disease; ln, natural log transformation; iFGF23, intact fibroblast growth factor 23; cFGF23, c-terminal fibroblast growth factor 23; DKK1, Dickkopf-1; Kt/V, urea kinetics; iPTH, intact parathyroid hormone; Alk-P, alkaline phosphatase; CI, confidence interval. *p* values less than 0.05 are marked in bold.

**Table 3 jcm-12-01550-t003:** Linear regression analysis of the association between iFGF23 and cFGF23 and bone mineral density in patients on chronic hemodialysis.

	BMD of Lumbar Spine		BMD of Femoral Neck	
	Estimate (95% CI)	*p*-Value	Estimate (95% CI)	*p*-Value
ln(iFGF23)				
Crude	−0.09 (−0.16, −0.02)	**0.015**	−0.03 (−0.09, 0.03)	0.271
Model 1	−0.12 (−0.19, −0.04)	**0.003**	−0.06 (−0.12, −0.01)	**0.028**
Model 2	−0.11 (−0.18, −0.03)	**0.006**	−0.06 (−0.12, −0.01)	**0.032**
Model 3	−0.10 (−0.18, −0.02)	**0.012**	−0.06 (−0.12, −0.003)	**0.037**
ln(cFGF23)				
Crude	−0.06 (−0.16, 0.04)	0.226	0.00 (−0.08, 0.08)	0.980
Model 1	−0.08 (−0.18, 0.03)	0.145	−0.03 (−0.11, 0.04)	0.369
Model 2	−0.07 (−0.18, 0.04)	0.184	−0.04 (−0.12, 0.04)	0.343
Model 3	−0.05 (−0.17, 0.07)	0.387	−0.04 (−0.13, 0.06)	0.430

Multivariable model 1 was adjusted for age, gender and hemodialysis duration. Multivariable model 2 comprised model 1 and diabetes mellitus, cardiovascular disease and body mass index. Multivariable model 3 was based on model 2 with the addition of ln(sclerostin), creatinine, albumin, ln(triglyceride) and Kt/V into the analysis. Abbreviations: ln, natural log transformation; iFGF23, intact fibroblast growth factor 23; cFGF23, c-terminal fibroblast growth factor 23; BMD, bone mineral density; CI, confidence interval; Kt/V, urea kinetic. *p* values less than 0.05 are marked in bold.

**Table 4 jcm-12-01550-t004:** Literature review of the association between serum FGF23 levels and BMD in chronic hemodialysis patients.

Study	Year	Number	Study Design	FGF23 Type *	BMD Method	Parameter Adjusting	Results
Urena Torres, P., et al. [32]	2008	99	Cross-sectional	cFGF23	DXA	No	cFGF23 was not correlated with BMD.
Park, S.-Y., et al. [33]	2010	54	Cross-sectional	iFGF23	DXA	Yes	iFGF23 was not correlated with BMD.
Malluche, H., et al. [28]	2014	81	Prospective	cFGF23	QTC	Yes	cFGF23 was a predictive factor of lower BMD of the spine.
					DXA		
Wu, Q., et al. [29]	2014	64	Cross-sectional	iFGF23	DXA	No	iFGF23 was positively associated with osteopenia and osteoporosis.
Zheng, S., et al. [34]	2018	125	Cross-sectional	iFGF23	DXA	Yes	iFGF23 was not associated with osteopenia and osteoporosis.
Bouksila, M., et al. [30]	2019	100	Cross-sectional	iFGF23	DXA	No	iFGF23 was positively associated with osteopenia and osteoporosis.
Slouma, M., et al. [31]	2020	90	Cross-sectional	iFGF23	DXA	No	iFGF23 was associated with osteopenia and osteoporosis of lumbar spine.
Our study	2021	43	Cross-sectional	iFGF23	DXA	Yes	iFGF23 had a negative correlation with BMD levels of both lumbar spine and femoral neck.
				cFGF23			

* If the method in the study did not mention the type of FGF23, we categorized the types by unit of FGF23 used in the study (pg/mL indicates iFGF23 and RU/mL indicates cFGF23). Abbreviations: FGF23, fibroblast growth factor 23; cFGF23, c-terminal fibroblast growth factor 23; iFGF23, intact fibroblast growth factor 23; BMD, bone mineral density; QTC: quantitative computed tomography; DXA: dual-energy X-ray absorptiometry.

## Data Availability

Supporting data can only be made available to bona fide researchers subject to a non-disclosure agreement due to confidentiality agreements. The data details and request access method are available from the corresponding author.

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
