# Peer review of "Serum Intact Fibroblast Growth Factor 23 Levels Are Negatively Associated with Bone Mineral Density in Chronic Hemodialysis Patients"

_jcm, 2023, doi:10.3390/jcm12041550_

Round 1

Reviewer 1 Report

Authors present a very interesting and up to date topic, involving the impact of Serum Intact Fibroblast Growth Factor 23 Levels in CKD-MBD, especially regarding Bone Mineral Density. As stated in the introduction, there are several similar studies performed in the recent years, but their results are discordant.  
I consider the author`s work as a clean study, well prepared and presented in a well-structured manner. The experimental design is appropriate for the purposes chosen by the authors. Although I have not the competence to review the statistical work, I appreciate that the results are presented in a transparent manner, The drawbacks are presented accurate and clear. There are coherent conclusions drawn and they are supported by the results, though the subject remains in debate and requires further studies.
The language is appropriate and understandable, although there are some phrases to review (e.g in the Abstract, the sentence before the last one is “iFGF23 also had a negative associated with BMD of the femoral neck after the multivariable models adjustment.”    

Author Response

Please see the attachment, thank you!

Reviewer 2 Report

The authors investigated the association between FGF23 and bone mineral density.

The topic is interesting, however, there are a few concerns in this manuscript.

According to the authors, cFGF23 has greater bioavailability and stability than iFGF23. However, the molecular weight of cFGF23 is lower than that of FGF23. Therefore, some amount of cFGF23 was removed by hemodialysis. I want to know the correlations between KT/V and cFGF23 and iFGF23.

It is said that lumber BMD in patients undergoing hemodialysis is not so accurate compared to femoral neck BMD in patients undergoing hemodialysis. This is because lumber BMD is affected by ectopic calcification or calcification of the vascular system. This study could not show that significant statistical association between FGF23s and femoral neck BMD. Please discuss this fact more. 

The length of the abstract is too long. According to the author guidelines of this journal, the abstract should be a total of about 200 words maximum.

Minor point

Reference 28 was retracted due to inconsistencies in the data.

Therefore, this reference should be excluded.

Author Response

Please see the attachment, thank you!

Round 2

Reviewer 2 Report

The manuscript is properly corrected. I do not have any comments now.